

# Modulating pain thresholds through classical conditioning

Juliane Traxler[1,2], Victoria J. Madden[1,3], G. Lorimer Moseley[4] and Johan W.S. Vlaeyen[1,2]

[1] Research Centre for Health Psychology, KU Leuven, Leuven, Belgium
[2] Experimental Health Psychology, Maastricht University, Maastricht, Netherlands
[3] Department of Psychiatry and Mental Health, Department of Anaesthesia and Perioperative Medicine, University of Cape Town, Cape Town, South Africa
[4] Body in Mind Research Group, University of South Australia, Adelaide, SA, Australia

## ABSTRACT

**Background:** Classical conditioning has frequently been shown to be capable of evoking fear of pain and avoidance behavior in the context of chronic pain. However, whether pain itself can be conditioned has rarely been investigated and remains a matter of debate. Therefore, the present study investigated whether pain threshold ratings can be modified by the presence of conditioned non-nociceptive sensory stimuli in healthy participant.

**Methods:** In 51 healthy volunteers, pain threshold to electrocutaneous stimuli was determined prior to participation in a simultaneous conditioning paradigm. Participants underwent an acquisition phase in which one non-painful vibrotactile stimulus (CS$^+$) was repeatedly paired with a painful electrocutaneous stimulus, whereas a second vibrotactile stimulus of the same quality and intensity (CS$^-$) was paired with a non-painful electrocutaneous stimulus. Stimulation was provided on the lower back with close proximity between the conditioned stimulus and the unconditioned stimulus. In the test phase, electrocutaneous stimuli at the individually-set threshold intensity were simultaneously delivered together with either a CS$^+$ or CS$^-$. Pain intensity ratings were obtained after each trial; expectancy ratings were obtained after each block. The primary outcome was the percentage of test stimuli that were rated as painful.

**Results:** Test stimuli were more likely to be rated as painful when they were paired with the CS$^+$ than when they were paired with the CS$^-$. This effect was not influenced by contingency awareness, nor by expectancies or mood states.

**Discussion:** The findings support the notion that the judgement of an event being painful or non-painful can be influenced by classical conditioning and corroborate the possible role of associative learning in the development and maintenance of chronic pain.

# INTRODUCTION

The development of chronic pain, that is, the persistence of pain long after tissue healing, is not well understood. A proposed mechanism in the transition from acute to chronic

Corresponding author
Juliane Traxler,
juliane.traxler@kuleuven.be

pain that seems intuitive and is widely accepted among clinicians (*Madden & Moseley, 2016*) is associative learning in the form of classical conditioning. It is assumed that during acute pain, nociceptive signaling serves as the unconditioned stimulus (US), and occurs alongside non-nociceptive signaling such as touch, proprioception and haptic perception, which serves as the conditioned stimulus (CS), such that the association of both stimuli is stored in memory. According to this model, the CS then carries information about the occurrence of the US: a stimulus repeatedly paired with pain (CS$^+$) indicates danger and elicits the expectancy of pain, while a stimulus paired with non-painful sensations (CS$^-$) indicates safety (*Domjan, 2005*). By virtue of the CS–US association, CS's may then instil conditioned responses (CR) such as increases in muscle tension, sympathetic activation, avoidance behavior or fear in the absence of the US (*Linton, Melin & Gotestam, 1984*). Some theorists have espoused the idea that pain itself can occur as a CR which continues to maintain itself after the initial source of nociception has subsided (*Flor & Birbaumer, 1994*; *Linton, Melin & Gotestam, 1984*; *Moseley & Vlaeyen, 2015*). Accordingly, classical conditioning might evoke and maintain chronic pain.

While it has repeatedly been shown that various pain-related constructs including expectancies, pain-related fear, fear of movement and avoidance behavior can be classically conditioned (*Claes, Vlaeyen & Crombez, 2016*; *Gramsch et al., 2014*; *Jepma & Wager, 2015*; *Meulders, Vansteenwegen & Vlaeyen, 2011*; *Peuterl et al., 2011*; *Zaman et al., 2015*), less is known about whether pain itself can be conditioned as well. Only a few attempts have been made to elicit pain as a classically CR (for a review, see *Madden et al., 2015*). So far, most studies evaluated *conditioned hyperalgesia* (increased painful response), and provided strong support that pain can indeed be amplified through associative learning, as identical nociceptive stimuli were found to evoke different perceptions merely depending on established, frequently unconscious associations (*Harvie et al., 2016a*; *Jensen et al., 2015*). In contrast, only three studies indirectly investigated *conditioned allodynia* (a painful response upon non-nociceptive stimuli) and reported contradictory findings (*Bräscher et al., 2017*; *Klinger et al., 2010*; *Williams & Rhudy, 2007*). Examining this topic further, Madden and colleagues conducted two targeted experiments using laser-stimuli (*Madden et al., 2016a*) or thermal stimuli (*Madden et al., 2016b*), both of which activate nociceptive Aδ- and C-fibers, as US, and vibrotactile stimuli activating non-nociceptive Aβ-fibers (*Abraira & Ginty, 2013*) as CSs, to specifically test the effect of classical conditioning on pain threshold ratings. At-pain-threshold stimuli are considered to be ambiguous, meaning they are perceived as painful or non-painful at chance level. It was hypothesized that the classical conditioning procedure would lead to a higher percentage of these at-pain-threshold stimuli being perceived as painful, signifying a decrease in pain threshold. However, only the former study (*Madden et al., 2016a*) demonstrated a decrease in pain thresholds on CS$^+$ trials whereas the latter (*Madden et al., 2016b*) did not. It was speculated that these inconsistent findings could be due to the different types of USs used, as laser stimuli produce more discrete sensations than contact heat stimuli, allow for faster temperature increases, and selectively activate Aδ- and C-fiber nociceptors whereas thermodes used for thermal stimulation concomitantly activate low threshold mechanoreceptors (*Plaghki & Mouraux, 2003*). Consequently,

studies on the conditioning of allodynia and pain thresholds are scarce and their findings are mixed at best. In order to better understand and extend previous findings, further research is required.

The present study aimed at examining whether the judgment of stimuli as painful or non-painful can be a classically CR in healthy participants by replicating and extending the studies by *Madden et al. (2016a, 2016b)*. Similar to these studies, in the present experiment two neutral vibrotactile stimuli ($CS^+$, $CS^-$) were paired with a painful ($US_P$) and a non-painful ($US_{NP}$) electrocutaneous stimulus, respectively, all of which were applied to the lower back. The USs evoked an unconditioned response (URs), namely reported pain. Subsequently, both $CS^+$ and $CS^-$ were presented simultaneously with at-pain-threshold electrocutaneous stimuli ($US_{PT}$). It was expected that these $US_{PT}$s would be experienced and rated as painful as opposed to non-painful (CR) on more of the $CS^+$ trials than the $CS^-$ trials, confirming that the judgment of stimuli as painful or non-painful can be modulated through classical conditioning. The main difference with the previous studies was the type of US used, namely an electrocutaneous stimulus instead of laser stimulation (*Madden et al., 2016a*) or thermal stimulation (*Madden et al., 2016b*). This type of stimulation was chosen as it shares some of the features of laser stimuli, namely the discreteness and speed with which activation of nociceptors is achieved, while co-activating mechanoreceptors like thermal stimuli. Thus, although not directly comparing the different types of USs, the present study may help to interpret previous contradictory findings.

## MATERIALS AND METHODS

### Participants

As the present experiment closely resembled the study by *Madden et al. (2016a)*, an a priori sample size calculation using G*Power (*Faul et al., 2007*) was conducted with the effect size obtained by *Madden et al. (2016a)* (ES = 0.58), according to which 34 participants would need to be included for a power of .95 ($\alpha = 0.05$). In order to deal with a potentially overestimated effect size of the original study (*Brandt et al., 2014*), the study aimed at including 50 participants in total. Participants were recruited at the University of Leuven (KU Leuven), Belgium, using flyers, posters, word of mouth, and through an online platform for participant recruitment.

Participants were eligible for participation if they were older than 18 years, proficient in the Dutch language and able to consent autonomously. Exclusion criteria based on self-report were (a) a history of chronic pain (defined as pain on most days for 3 months or longer), (b) acute pain at the time of testing, (c) use of analgesic medication on the day of testing, (d) use of medication that could alter skin sensitivity or healing, (e) a skin condition that would not allow to tolerate electrode application without damage (e.g., a past operation at the location where the electrodes would be placed), (f) sensation problems, (g) a serious medical condition (e.g., diabetes mellitus, peripheral vascular disease, cardiovascular disease, neurological problems), (h) pregnancy, (i) use of an electronic implant (e.g., a pacemaker) or (j) a previous or current psychiatric diagnosis. Furthermore, participants were excluded mid-procedure if they (1) were unable to

discriminate the locations of the vibrotactile stimuli during baseline (<50% correct), and/or (2) reported a pain threshold that was too high or too low for the equipment's technical and safety settings.

Written informed consent was obtained from each participant prior to testing, and participation was rewarded with course credit or with the chance of winning a weekend trip worth 250 Euros. The procedures conformed to the Helsinki Declaration and were approved by the institutional ethics committee (Social and Societal Ethics Committee; approval reference number: G-2016 11 663).

## Stimuli

Electrocutaneous stimuli were used for the unconditioned stimuli. They were delivered by means of two reusable Coulbourn electrodes (four mm) that were attached five mm apart from each other on participants' lower back and were controlled by a constant-current stimulator (DS7a; Digitimer Limited, Hertfordshire, UK). Single pulses as well as 3-pulse, 5-pulse and 7-pulse trains were used, with a duration of 0.5 ms each, and an inter-stimulus interval of five ms. Using pulse trains allows for the production of more stable stimulus intensities (*Mouraux et al., 2013*; *Mouraux, Marot & Legrain, 2014*). Due to this effect, a lower electric current was necessary to evoke a painful sensation, which improved the signal-to-noise ratio and reduced the risk of causing burns to the skin (*Inui et al., 2002*). Stimulus intensities were determined by first determining individual pain thresholds using a 5-pulse train. This was called the pain-threshold intensity. The non-painful stimuli ($US_{NP}$) were single pulse stimuli at the pain-threshold intensity and the painful stimuli ($US_P$) were 7-pulse trains at the pain-threshold intensity. During the test phase, 3-pulse trains served as at-pain-threshold stimuli ($US_{PT}$) instead of 5-pulse trains in order to account for sensitization and for perceived control during the calibration phase as participants calibrated the pain threshold intensity themselves by means of clicking on "more" or "less" buttons displayed on the computer screen. The output of this was visible for the experimenter who adjusted the stimulus intensity on the DS7a accordingly. All choices about pulse-trains and calibration procedure were based on a pilot study that is described in the section *Procedure*.

Vibrotactile stimuli were used as CSs and were delivered by three tactors taped to the skin on the left side of the lower back. One tactor (cephalad/"upper") was placed 40 mm above the electrodes, one (caudad/"lower") was placed 40 mm below the electrodes, and one was attached 15 mm lateral to the electrodes (see Fig. 1). The intensity of vibrations was set to a clearly perceptible but non-painful level, which remained the same for all participants, with a duration of 17 ms. Either the caudad or cephalad tactor was conditioned to be CS+ by repeatedly being paired with a painful electrocutaneous stimulus ($US_P$), while the other became the CS− by being paired with a non-painful electrocutaneous stimulus ($US_{NP}$). The onset of the two stimuli (electrocutaneous + vibrotactile) occurred simultaneously at all times. Allocation of tactor to CS+ or CS− was counterbalanced across participants in a randomized order. The lateral tactor was only used for calibration purposes and remained at that (neutral) location during the whole experiment.

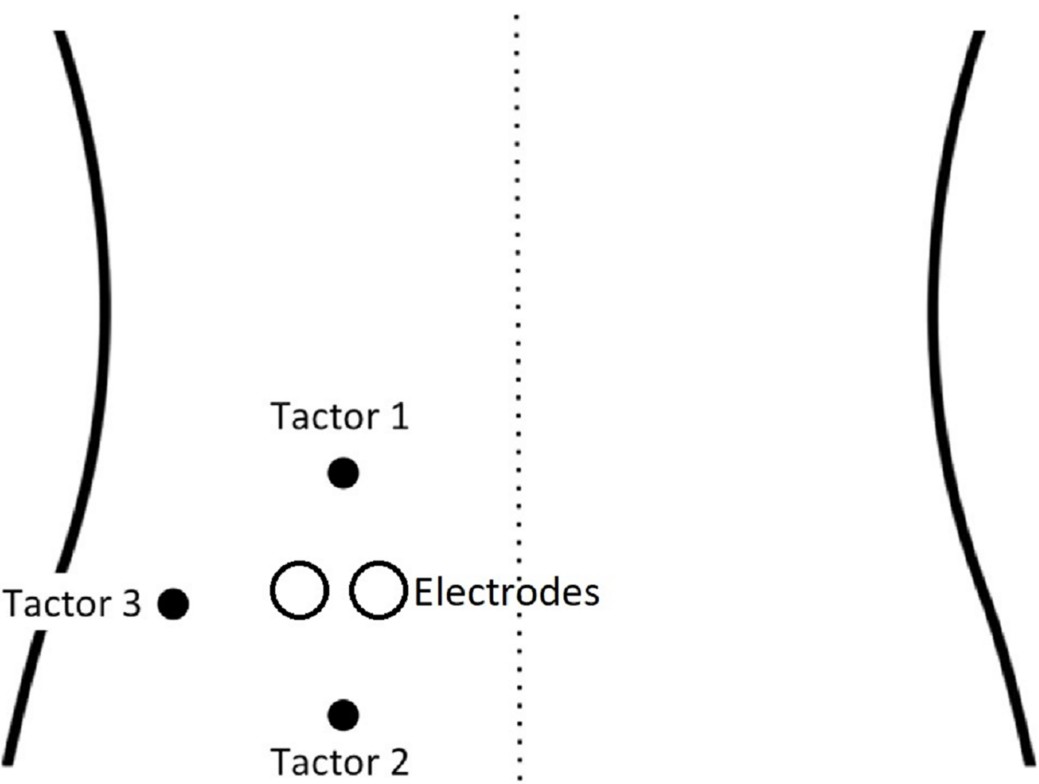

**Figure 1 Set-up of vibrotactors and the electrodes on the back (adapted with permission from *Madden et al. (2016a)*).**

## Measures and outcomes

### Manipulation check

In order to check whether the conditioning procedure was successful, US expectancy ratings were conducted at several time points during the study (see Procedure). Participants were required to answer the questions (i) "*To what extent do you expect the stimulus package to be painful (rather than non-painful) if the stimulus package includes the upper vibration?*" and "*To what extent do you expect the stimulus package to be painful (rather than non-painful) if the stimulus package includes the lower vibration?*" on a numerical rating scale ranging from 0 = "I do not expect that it will feel painful" to 10 = "I fully expect that it will feel painful."

### Positive and negative affect

The Positive and Negative Affect Schedule (PANAS; *Watson, Clark & Tellegen, 1988*) is a self-report measure assessing positive and negative affect. Participants indicate the extent to which 20 different emotional states (e.g., guilty, inspired, restless) are currently applicable to them on a 5-point Likert scale (1 = not at all, 5 = very much). The two subscales have good psychometric properties, including good convergent and discriminant validity (*Crawford & Henry, 2004*). In the present sample, there was a high internal consistency for the positive affect subscale ($\alpha = 0.88$). However, that for the negative affect subscale was rather low ($\alpha = 0.60$).
### Participant perceptions

At the end of the experiment, participants answered a number of questions in order to assess (1) whether they were naïve to the purpose of the experiment (i.e., blinding check), (2) whether they perceived differences between the vibrations of the two CS tactors, (3) how they perceived the timing of the vibrations relative to the electrocutaneous stimuli, (4) whether they noticed a relationship between the vibrotactile CSs and the painfulness of the electrocutaneous USs (i.e., contingency awareness/awareness of the conditioning paradigm), and (5) whether they experienced a direct association between their adaptation of the stimulus intensity and the actual intensity during the calibration phases.

### Outcome

The primary outcome of this study was the percentage of trials during test phase in which at-pain-threshold electrocutaneous stimuli ($US_{PT}$ at test phase) were experienced as painful, when the $CS^+$ was active, as compared to when the $CS^-$ was active. For that purpose, participants indicated whether or not each "stimulus package," that is, the combination of either vibration with the electrocutaneous stimulation, was painful or non-painful and rated the intensity of the stimuli on the sensation and pain rating scale (SPARS, initially called the FESTNRS; *Madden et al., 2016b*) within a maximum of 15 s after stimulus onset. The SPARS captures both painful and non-painful experiences as it runs from −50 = "no sensation," through 0 = pain threshold ("the exact point at which you feel transition to pain") to +50 = "most intense pain you can imagine." Participant ratings on the SPARS show a curvilinear relationship to stimulus intensity, with a slightly steeper slope at the extremes of the scale and stable stimulus-response characteristics across the range of the scale (*Madden et al., in press*). A visual scale showing the range and the anchors was presented on a computer screen on which participants could select any number within this range. Furthermore, participants reported which tactor they felt (upper vs. lower) during each trial by clicking one of two buttons.

## Procedure

The procedure of this study was adapted from *Madden et al. (2016a)*. The study has a within-subject design and consists of four main phases (see also Fig. 2). The protocol has been locked on the online database Augias from KU Leuven from which we did not deviate.

### Pilot study

A pilot study with 17 participants was conducted in order to test and optimize the methods and procedures to be used in the full-scale experiment. The general procedure did not have to be changed and can be found below. However, the ideal number of pulse trains for delivering non-painful, at-pain-threshold and painful stimuli had to be tested resulting in a 1-4-7 grading, which provided easily differentiable intensities.

Importantly, pain thresholds proved very unstable, with both habituation and sensitization occurring throughout the experiment. In order to counteract this, recalibration phases were included between acquisition and test blocks. Regarding sensitization, participants were asked to (re-)calibrate to −5 instead of zero so as to ensure that the stimulus

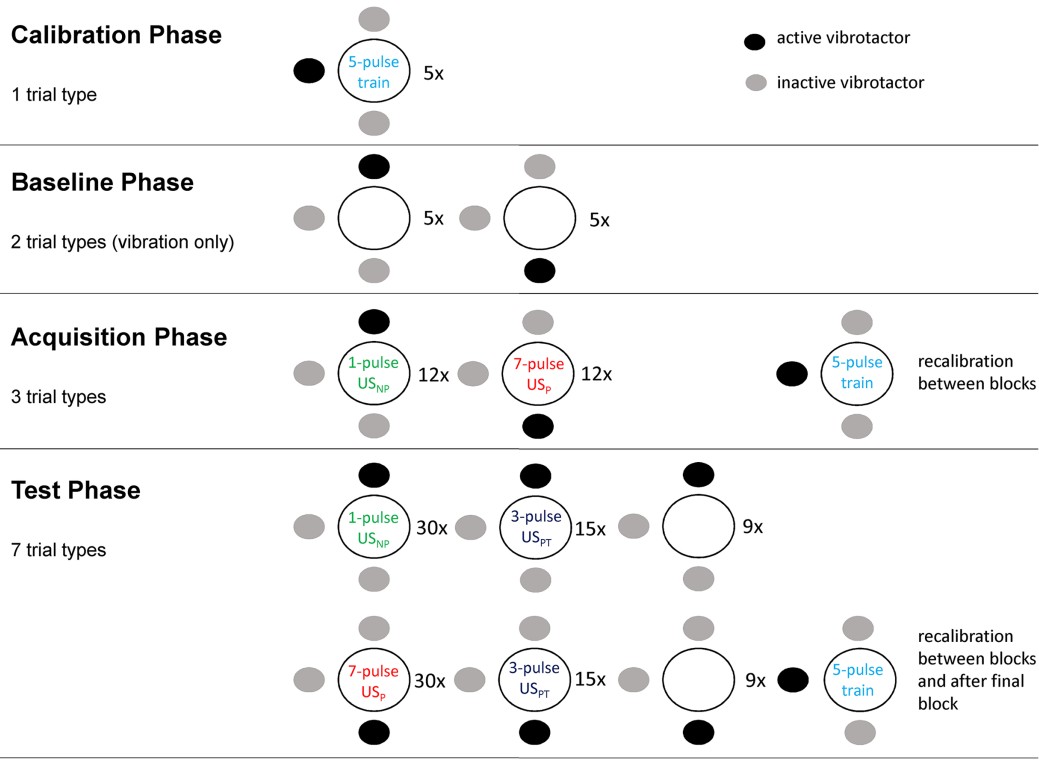

**Figure 2 The four phases of the experimental procedure.** Dots indicate vibrotactile stimuli, dark dots show which vibrotactor was activated for each trial type. Circles indicate the field on which the two electrodes were placed (adapted with permission from *Madden et al. (2016a)*).

would be closer to just non-painful rather than to just painful. Additionally, it was decided to use a 3-pulse train as the at-pain-threshold stimulus ($US_{PT}$) in the test phase because the 5-pulse train which had been calibrated to −5 was consistently rated as painful during testing. We also tried to minimize context changes between (re-)calibration phases and test phases. However, given the frequent (re-)calibrations, a fast and precise calibration procedure was needed. The pilot study showed that self-adjustment through the participant by means of clicking "more" and "less" buttons displayed on a computer screen was more efficient than the commonly used adaptive staircase procedure (*Madden et al., 2016a*).

A final aspect that was addressed in the pilot study was the intensity and distance between the tactors. The distance between the tactors was increased from 70 (*Madden et al., 2016a*) to 80 mm and the vibration intensity was calibrated to a level that would permit good distinguishability.

### Preparation

Prior to testing, participants received written information about the study without details about the study goals and exact procedures so as to ensure blinding. Participants were screened for exclusion criteria via e-mail. Upon arrival at the lab, participants once more received general study information. They had the opportunity to ask questions and

were instructed to carefully read and sign the informed consent form and an exclusion criteria form afterward. Participants were seated in front of a computer, straddling a chair. Before the electrodes were attached on the left side of the lower back, participants crossed their arms over their chest, looked upward, and bent backward. The line of greatest curvature or "hinge" was identified. Moving along this line, the two electrodes were taped to the skin 40 mm left of the spine once the participant had resumed a comfortable resting posture. The tactors were placed 40 mm above, 40 mm below and 15 mm to the left of the electrodes.

### Calibration phase

In this phase, participants' pain thresholds for the electrocutaneous stimulus (5-pulse train) were calibrated and participants could practice reporting on the SPARS. First, they received a 0.1 mA stimulus paired with the lateral tactor as a neutral vibrotactile stimulus and were asked to rate their experience. Next, they were instructed to adjust the stimulus intensity to match their personal ratings of −30, −10, +10, +30 and finally zero (threshold) by clicking on "more" or "less" buttons displayed on the computer screen. All electrocutaneous stimuli in the calibration phase were delivered simultaneously with the lateral tactor. Next, participants received and rated six electrode-tactor combinations with intensities between those previously determined as −30 and +30, and again adjusted the intensity to their threshold and finally to −5. This last intensity served as the pain threshold so as to ensure that it was closer to just non-painful rather than to just painful (to account for anticipated perceptual sensitization) and was subsequently used for all single pulses and pulse trains.

### Baseline phase

Before the conditioning procedure, baseline ratings of the CSs were obtained. Participants received 10 vibration-only trials, that is, the upper and lower tactor were activated five times each, rated each trial on the SPARS and indicated which tactor was activated. Participants who were unable to discriminate the tactor locations (<50% correct) were excluded at this point in the procedure. Subsequently, participants provided expectancy ratings.

### Acquisition phase

During the acquisition phase, participants received 12 painful electrocutaneous stimuli (7-pulse trains—$US_P$) paired with the tactor that was assigned to become the $CS^+$. Similarly, the other tactor ($CS^-$) was paired with the non-painful electrocutaneous stimulus (single pulses—$US_{NP}$). The trials were delivered in three blocks of eight trials each and were pseudo-randomly ordered with no more than two identical trials delivered successively. The reinforcement contingency was set to 100%. Participants were instructed to rate each stimulus package on the SPARS. After each block, participants were asked to give expectancy ratings and to readjust the intensity of the 5-pulse trains (in combination with the lateral tactor) to −5 in order to counteract habituation and sensitization. These newly determined at-pain-threshold intensities were then used as stimulus intensity in the subsequent block.

### Test phase

In total, the test phase consisted of three blocks of 36 trials each (108 trials in total) with the following stimulus packages being delivered: 15 $CS^+$/$US_{PT}$ trials and 15 $CS^-$/$US_{PT}$ trials as main outcomes, 30 $CS^+$/$US_P$ trials and 30 $CS^-$/$US_{NP}$ trials as reinforcement in order to prevent extinction, and nine $CS^+$-only trials and nine $CS^-$-only trials. These stimulus packages were delivered in random order with no more than two identical trials being delivered successively. After each trial participants provided intensity ratings on the SPARS and indicated the tactor location. Again, they gave expectancy ratings and readjusted the intensity of the 5-pulse trains to −5 between blocks. Furthermore, after the final test block, participants were instructed to adjust the intensity of the 5-pulse train to their pain threshold (zero on SPARS) two times, one time in combination with the upper tactor and the other time with the lower tactor. Whether recalibration was done with the upper or the lower tactor first was counterbalanced across participants.

Finally, participants filled in the PANAS and the five questions about their perceptions of the study. After this, the electrodes and vibrotactors were removed. Participants were debriefed, assigned their course credit and thanked for their participation.

### Statistical analysis

Data were analysed using SPSS 25.0 (IBM SPSS Statistics for Windows, Version 25.0; Armonk, NY, USA). Plots were generated in R v3.4.3 (*R Core Team, 2017*), in Rstudio v1.1.414 (*RStudio Team, 2018*). The following packages were used: readr (*Wickham et al., 2017*), magrittr (*Bache & Wickham, 2014*), tidyverse (*Wickham, 2017*), and ggplot2 (*Wickham, 2009*).

Prior to analysis, data were checked for normality, sphericity, outliers and missing values. The only missing values were found for the indication of tactor localization. These were omitted for the analyses on tactor localization mistakes. When appropriate, non-parametric tests were applied. Follow-up pairwise comparisons with Bonferroni adjustment were applied to ANOVAs with significant results.

To rule out any baseline differences in vibration intensities of the two tactors, a Wilcoxon signed rank test was conducted on baseline intensity ratings, with Condition ($CS^+$ vs. $CS^-$) as within-subject factor. To test whether manipulations were successful, expectancy ratings as well as intensity ratings were analyzed by means of two separate 2 (Condition: $CS^+$ vs. $CS^-$) × 3 (Phase: baseline, acquisition, test) repeated measures (RM) ANOVAs. Additionally, blinding for the study goal, contingency awareness for the CS–US pairings, perceived control over the stimulus calibrations as well as quality of the vibrotactile stimuli and simultaneity of the vibrotactile and electrocutaneous stimuli were checked.

Subsequently, in order to test the main hypothesis, a binary variable reflecting non-painful ($\leq 0$) and painful ($>0$) at-pain-threshold trials was computed, based on which a second variable reflecting the percentage of $US_{PT}$ trials rated as painful was created. Next, the percentage of $US_{PT}$ trials rated as painful was compared across the two conditions, that is, between the 15 $CS^+$/$US_{PT}$ and 15 $CS^-$/$US_{PT}$ stimulus packages, by means of a Wilcoxon signed rank test—this was the primary analysis. As secondary analyses, a 2 (Condition: $CS^+$/$US_{PT}$ vs. $CS^-$/$US_{PT}$) × 3 (Test Block: 1, 2, 3) RM ANOVA

was conducted to examine time effects, and a Wilcoxon signed rank test was conducted to compare the intensity ratings for the $CS^+/US_{PT}$ and $CS^-/US_{PT}$ trials. As additional measures of shifts in perception, two Wilcoxon signed rank tests were used to (1) compare the intensity ratings for the $CS^+$-only and $CS^-$-only trials and (2) compare the recalibrated at-pain-threshold intensities (from the final re-calibration phase) when paired with the $CS^+$ compared to $CS^-$. For the latter analysis, the recalibrated intensities for the $CS^+$ and $CS^-$ were divided by the respective group mean recalibrated intensities. In order to control for mistakes on the tactor localization task, these analyses were repeated with perceived rather than actual tactor location, that is, trials on which the $CS^+$ location was wrongly perceived as the $CS^-$ location were considered $CS^-$ trials in the analyses, and vice versa. Lastly, the relationship between the primary outcome and negative affect (PANAS), contingency awareness, and expectancy, were explored by means of analyses of covariance (ANCOVAs), with total scores of the positive and negative subscales of the PANAS, the binary variable contingency awareness (yes/no) and the expectancy ratings given for the upper and lower tactor during test phase.

## RESULTS

A total of 51 participants (seven males) aged 18–31 years ($M = 19.39$, SD = 2.55) were included. Two participants were excluded mid-procedure for having no clear non-painful range and perceiving almost all of the stimuli as painful. Four other participants were excluded due to technical problems that may have rendered their data unreliable. None were excluded for inability to discriminate between the vibrotactile stimuli. Consequently, the analyses were conducted on the final sample of 45 participants (six males). Participant characteristics are described in Table 1. The mean intensity of at-pain-threshold stimuli at the initial calibration was 3.18 mA (SD = 1.945).

### Blinding and contingency awareness

At the completion of the experiment, 40/45 participants were unaware of the goal of the study and 17 were unaware of the contingency, that is, either they reported to have noticed no correlation at all between the location of the vibration and whether the electrocutaneous stimulus was painful or not, or they identified the $CS^-$ to be associated with the painful stimuli. Fourteen participants experienced differences between the vibrations from the upper and the lower tactor, 10 of whom perceived the $CS^+$ as "stronger," "sharper," and "longer" and four of whom did not specify the perceived difference. Four participants indicated that the US and CS did not have a simultaneous onset but that they perceived either the vibrotactors ($n = 3$) or the electrocutaneous stimulus ($n = 1$) first. Lastly, 29 participants reported that they experienced control over the calibration whereas 16 reported that they did not.

### Manipulation checks

There was initially no significant difference between the intensity ratings for the two vibrotactors at baseline (Wilcoxon signed-rank test: $Z = -1.084$; $p = 0.278$; $CS^+ = -32.22$; $CS^- = -33.52$). Similarly, expectancy ratings at the end of baseline did not differ

**Table 1 Participant characteristics (_n_ = 45).**

| Outcome | Mean (SD) | Range |
|---|---|---|
| Age | 19.44 (2.668) | 18–31 |
| Positive state affect (PANAS) | 26.69 (6.657) | 14–39 (normal range)[1] |
| Negative state affect (PANAS) | 13.36 (2.647) | 10–19 (normal range)[1] |

Note:
[1] Clinical meaning of PANAS scores based on _Crawford & Henry (2004)_.

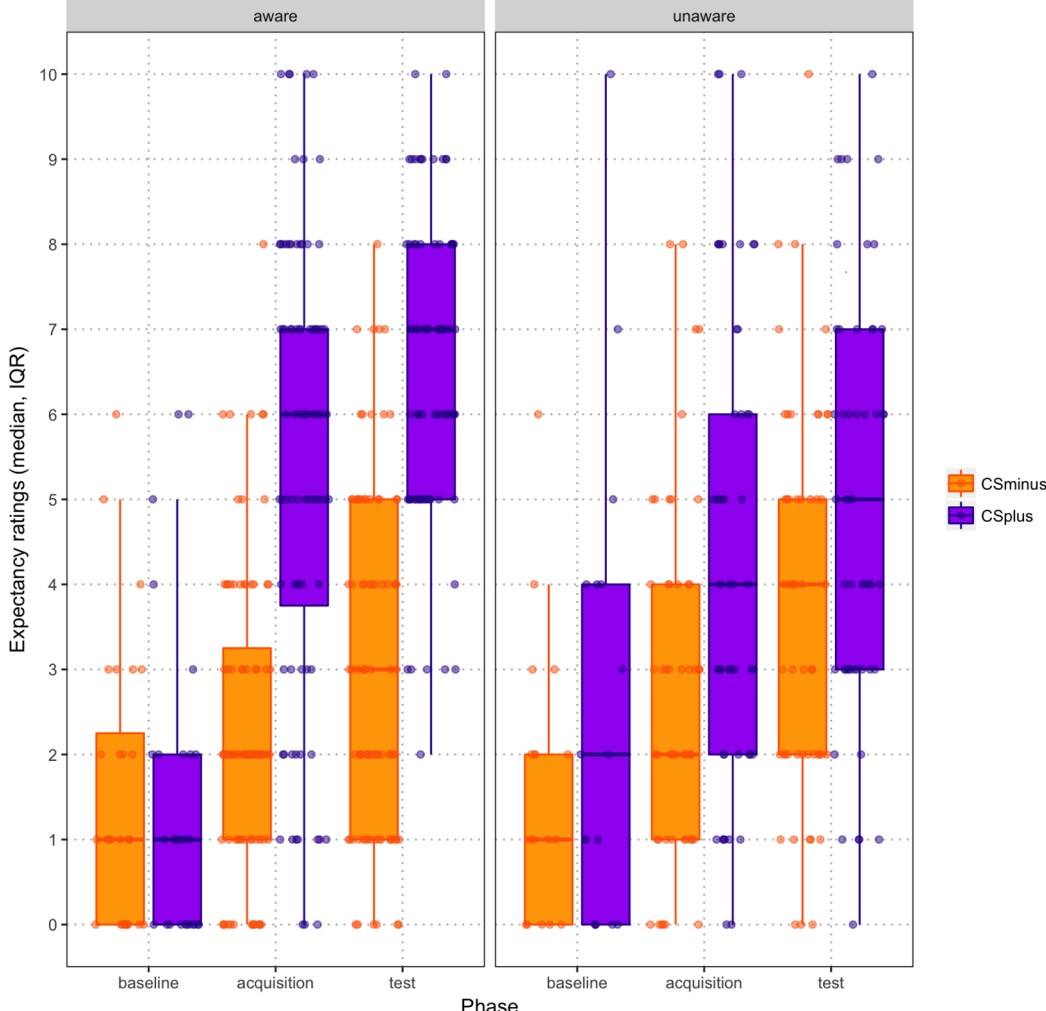

**Figure 3 Scatter boxplot of expectancy ratings by CS type across baseline, acquisition and test phases (mean, standard error), paneled by contingency awareness.**

significantly between the two vibrotactors (Wilcoxon signed-rank test: $Z = -1.302$; $p = 0.193$; $CS^+ = 1.93$; $CS^- = -1.51$). Expectancy ratings for $CS^+$ and $CS^-$ across the three phases are depicted in Fig. 3.

Across acquisition and test phase, $CS^+/US_P$ trials were rated as more intense than $CS^-/US_{NP}$ trials (main effect of Condition, $F(1,44) = 225.148$; $p < 0.001$; ($\eta_P^2 = 0.837$; 95% CI [23.062–30.218]; $CS^+/US_P$: $M = 9.630$, $SE = 1.024$; $CS^-/US_{NP}$: $M = -17.010$,

**Table 2 Mean stimulus intensity ratings (SD) for the three phases.**

|  | Baseline phase | Acquisition phase | Test phase |
|---|---|---|---|
| CS$^+$/US$_P$ |  | 3.722 (1.862) | 12.213 (1.562) |
| CS$^-$/US$_{NP}$ |  | −19.181 (2.859) | −13.528 (2.376) |
| CS$^+$/US$_{PT}$ |  |  | 6.174 (5.109) |
| CS$^-$/US$_{PT}$ |  |  | 3.596 (3.714) |
| CS$^+$ only | −32.811 (3.150) |  | −24.821 (15.464) |
| CS$^-$ only | −35.489 (2.822) |  | −27.222 (12.992) |

Note:
CS, conditioned stimulus; US$_P$, painful unconditioned stimulus; US$_{NP}$, non-painful unconditioned stimulus; US$_{PT}$, at-threshold unconditioned stimulus.

SE = 1.434) and this increase in intensity ratings was greater for CS$^+$/US$_P$ trials than for CS$^-$/US$_{NP}$ trials (Phase × Condition interaction, $F(1,44) = 11.901$, $p = 0.001$; $\eta_p^2 = 0.213$). Follow-up pairwise comparisons indicated that intensity ratings for CS$^+$/US$_P$ trials and CS$^-$/US$_{NP}$ trials increased from acquisition phase to test phase ($p < 0.001$).

In order to check whether the classical conditioning procedure was successful, a 2 (Condition: CS$^+$ vs. CS$^-$) × 3 (Phase: baseline, acquisition, test) RM ANOVA was performed on expectancy ratings. This analysis revealed a significant Condition × Phase interaction ($F(2,43) = 14.927$, $p < 0.001$; $\eta_p^2 = 0.410$): follow-up pairwise comparisons revealed that expectancy ratings for both CS$^+$ and CS$^-$ increased significantly from post-baseline to post-acquisition as well as from post-acquisition to post-test (both $p < 0.001$). Finally, participants who were contingency aware showed a greater increase over time in the difference between expectancy ratings for CS$^+$ and CS$^-$ (see Fig. 3; Condition × Phase × Contingency Awareness interaction, $F(2,42) = 9.372$, $p < 0.001$; $\eta_p^2 = 0.308$).

## Main hypothesis: effects on intensity ratings of pain threshold stimuli

Pain intensity ratings for CS$^+$/US$_P$ and CS$^-$/US$_{NP}$ trials, for CS$^+$/US$_{PT}$ and CS$^-$/US$_{PT}$ trials as well as for CS$^+$ only and CS$^-$ only trials are presented in Table 2. The test of the current study's main hypothesis showed that participants rated more US$_{PT}$ trials as painful when they were paired with the CS$^+$ (81.5%, SD = 21.03) than when they were paired with the CS$^-$ (72.7%, SD = 24.69; Wilcoxon signed rank test: $Z = -3.033$; $p = 0.002$). In absolute numbers, 12.22 (SD = 3.154) out of 15 CS$^+$/US$_{PT}$ trials were rated as painful compared to 10.91 (SD = 3.704) of the 15 CS$^-$/US$_{PT}$ trials. This difference was still apparent when tactor localization mistakes were controlled for (83.05% (SD = 19.22) of CS$^+$/US$_{PT}$ trials vs. 70.24% (SD = 26.29) of CS$^-$/US$_{PT}$ trials rated as painful, $Z = -3.768$; $p < 0.001$). The examination of time effects showed that participants rated more US$_{PT}$ trials as painful when they were paired with the CS+ than when they were paired with the CS- across all three blocks of the test phase (main effect of Condition, $F(1,44) = 10.201$; $p = 0.003$; $\eta_p^2 = 0.188$), and that this difference decreased over the three blocks of the test phase (main effect of Block ($F(2,43) = 3.360$; $p = 0.044$; $\eta_p^2 = 0.135$; no Block × Condition interaction, $F(2,43) = 0.332$; $p = 0.719$).

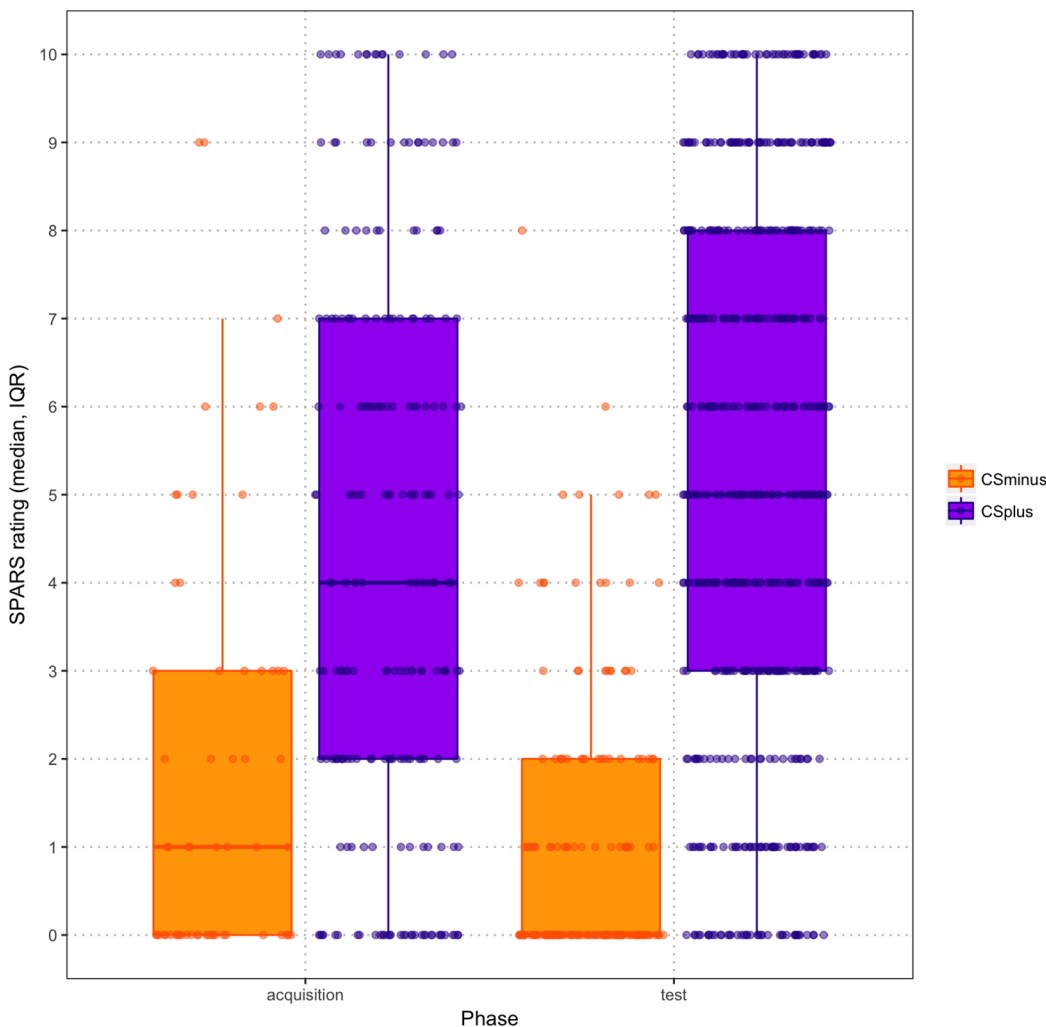

**Figure 4 Scatter boxplots of exact mean intensity ratings (error bars = standard error) by CS type during acquisition and test phase.**

## Secondary analyses: effects on intensity ratings

Participants rated $US_{PT}$ trials as significantly more painful when the $US_{PT}$ was paired with the $CS^+$ ($M = 6.585$, SD = 5.128) than when the $US_{PT}$ was paired with the $CS^-$ ($M = 4.113$, SD = 4.989; Wilcoxon signed rank test: $Z = -3.302$; $p = 0.001$). This difference was still apparent when tactor localization mistakes were controlled for ($Z = -3.515$; $p < 0.001$; $M_{CS+/UST} = 6.881$, SD = 4.556; $M_{CS-/UST} = 3.779$, SD = 5.703).

Exact mean intensity ratings for $CS^+/US_P$ and $CS^-/US_{NP}$ trials during acquisition and test phase are shown in Fig. 4. Overall, participants rated stimulus intensities higher in the test phase than in the acquisition phase. When examining the exact mean intensity ratings, a 2 (Condition: $CS^+/US_P$ vs. $CS^-/US_{NP}$) × 2 (Phase: acquisition, test) RM ANOVA showed that there was a significant Condition × Phase interaction effect ($F(1,44) = 11.901$; $p = 0.001$; $\eta_p^2 = 0.213$). Pairwise comparisons revealed that intensity ratings differed significantly between acquisition and test phase ($M_{acquisition} = -6.120$, SE = 1.159;

$M_{\text{test}} = -1.259$, SE = 0.874; $p < 0.001$) as well as between the vibrotactors ($M_{\text{CS+}} = 9.630$, SE = 1.024; $M_{\text{CS-}} = -17.01$, SE = 1.434; $p < 0.001$). None of these results changed notably when contingency awareness was added to the analysis as a covariate.

### Exploratory analyses

Intensity ratings for the $CS^+$-only and $CS^-$-only trials did not differ between the two conditions (Wilcoxon signed rank test: $Z = -1.155$; $p = 0.248$). This did not change when accounting for tactor localization mistakes on the CS-only trials ($Z = -1.307$; $p = 0.191$). Recalibrated stimulus intensities, that is, for the 5-pulse train electrocutaneous stimulus paired with the lateral vibrotactor, increased significantly from first (calibration phase) to last (before final test block) calibration moment ($Z = -5.842$; $p < 0.001$), with a mean intensity of 3.18 mA (SD = 1.945) at first calibration and 6.39 mA (SD = 4.716) at the last recalibration. However, final recalibrations for $CS^+$ and $CS^-$ did not differ ($Z = -0.035$; $p = 0.972$), with a mean intensity of 7.82 mA (SD = 5.045) for the $CS^+$ and 7.69 mA (SD = 5.014) for the $CS^-$.

The ANCOVAs revealed no effects of negative affect, contingency awareness, or expectancy ratings on the percentage of $CS^+$ and $CS^-$ trials rated as painful, on the intensity ratings of the $CS^+$-only and $CS^-$-only trials, or on the final recalibrations with $CS^+$ and $CS^-$ (all $p$-values > 0.05).

## DISCUSSION

The aim of the present study was to investigate whether simultaneous classical conditioning can modulate pain reports of at-pain-threshold stimuli. This knowledge may inform whether classical conditioning could be one of the mechanisms underlying the development or maintenance of chronic pain. It was expected that, after an acquisition phase in which non-painful vibrotactile stimuli (CSs) were repeatedly paired with either painful or non-painful electrocutaneous stimuli, $US_{\text{PT}}$s would be experienced as painful on more of the $CS^+$ trials than the $CS^-$ trials, confirming that pain threshold ratings can be modulated through classical conditioning.

The results confirmed our hypothesis: at-pain-threshold electrocutaneous stimuli evoked a CR, that is, they were 6% more likely to be reported as painful and were rated on average as more painful, if they were presented with $CS^+$ vibrotactile stimuli than if they were presented with the $CS^-$ vibrotactile stimuli. Besides, the exact pain ratings for $CS^+/US_{\text{PT}}$ trials were significantly higher than those for $CS^-/US_{\text{PT}}$ packages. Notably, the majority of $CS^-/US_{\text{PT}}$ trials were also perceived as painful. The calibration and classical conditioning procedures were successful, as indicated by significant increases in intensity ratings across all three phases as well as in pain expectancies from post-baseline to post-acquisition for the $CS^+$ but not the $CS^-$, even though many participants were not explicitly contingency aware. Neither the CS-only trials nor the final recalibrations were significantly affected by classical conditioning. Recalibrated pain threshold intensities increased throughout the experiment, indicating habituation.

The present findings are largely in line with the study by *Madden et al. (2016a)* and the study by *Williams & Rhudy (2007)*, in that exposure to non-nociceptive vibrotactile

stimuli was found to influence pain thresholds through classical conditioning. The results specifically parallel those by *Madden et al. (2016a)*, as even the difference between intensity ratings for $CS^+/US_{PT}$ and $CS^-/US_{PT}$ trials was roughly the same in both studies, while CS-only trials were unaffected by the conditioning procedure. Interestingly, despite the inclusion of re-calibration procedures, a higher percentage of at-pain-threshold trials, including 72% of $CS^-/US_{PT}$ trials, was perceived as painful in this study than in the study by *Madden et al. (2016a)*. Nevertheless, by finding comparable results while using different nociceptive stimuli (electrocutaneous vs. laser), this replication clearly strengthens the notion that classical conditioning can modulate judgements about whether an event is painful or not. Furthermore, both studies used a simultaneous conditioning paradigm in which CS and US were presented at the same time. This approach excludes expectation or changes in arousal—to which shifts in pain thresholds have been attributed in other classical conditioning studies (*Harvie et al., 2016a*; *Miguez, Laborda & Miller, 2014*)—as the mechanism of effect. Indeed, the simultaneous approach seems well suited to model chronic pain as the real-world CS (e.g., tactile cue) and US (nociceptive signaling) are likely to occur at the same time (*Madden et al., 2016b*), thus enhancing the validity of our findings. Finally, as half of our participants were not contingency aware, the present study also supports the claims by *Harvie et al. (2016a)* and *Jensen et al. (2015)* that expectations or conscious awareness of the learned association are not required for classically conditioned shifts in pain thresholds to occur.

Importantly, our findings contradict those by *Madden et al. (2016b)*, who did not find an effect of classical conditioning on pain thresholds. One difference between the two studies is the type of US, namely electrocutaneous vs. thermal stimuli. We specifically chose to use a different nociceptive stimulus than both studies by *Madden et al. (2016a, 2016b)*, not only in order to investigate whether the previous findings extend to other stimulus modalities, but also to help interpret their contradictory findings. The electrocutaneous stimulus in the present study is considered to be less discrete than laser stimuli but more discrete than contact heat stimuli (*Plaghki & Mouraux, 2003*). Previous investigations of conditioned fear of touch confirm that the spatial precision of somatosensory cues influences the likelihood of associative learning (*Harvie et al., 2016b*) and it seems possible that a similar mechanism is relevant here. The three stimulus types have different characteristics: laser stimuli isolate nociceptors (*Mouraux & Iannetti, 2009*) and therefore lack ecological validity, whereas the other two modalities stimulate both nociceptive and non-nociceptive mechanisms; thermal stimuli are more physiological than electrocutaneous stimuli, which activate mechanoreceptors as well as nociceptors but via their axons as well as receptors and nerve endings (*Handwerker & Kobal, 1993*), but chronic pain seldom develops after repeated exposure to noxious heat stimuli. This difference in discreteness between different types of nociceptive stimulation may also be a possible explanation for the diverging results given by *Madden et al. (2016b)*. One path forward that might overcome these limitations is to use proprioceptive stimuli (related to movement) as the conditioning stimuli (*Meulders, Vansteenwegen & Vlaeyen, 2011*).

The present study has several limitations that require attention. First of all, participants were frequently unsure about their indication of tactor location and many reported

difficulties at the end of the experiment. In these cases, some tended to base their choice of tactor location on whether the accompanying electrocutaneous stimulus was painful or not, which resulted in location-mistakes on 12.19% of all trials, on 15.85% of all $CS^+/US_{PT}$ trials and even 27.11% of all $CS^-/US_{PT}$ trials and which might indicate a reversed form of conditioning. That is, the simultaneous presentation of vibrotactile and electrocutaneous stimuli meant that each contained information about the occurrence of the other. Consequently, the electrocutaneous stimuli may have served as CS (providing information about the tactor) on some trials. A possible explanation for this error rate might be the close proximity of the tactors. We tried to improve discriminability between tactors by increasing the distance between them from 70 mm (as in the study by *Madden et al. (2016a)*) to 80 mm, which led to fewer mistakes than in the pilot study, yet the problem still occurred occasionally. A second point of criticism is that the vibrotactors made an audible sound that could have guided participants' attention to the tactor location. Nevertheless, we decided not to use white noise to mask it because of practical reasons: the calibration procedure was not compatible with continuous white noise delivered via headphones, and switching off the white noise during calibration would have formed an obvious change of context that could have interfered with conditioning. Moreover, white noise would have caused additional exhaustion of the participants. However, as mentioned previously, participants did not seem to be able to use the emitted sounds to locate the vibrotactors, as mistakes were made nonetheless. Another limitation is that not only most of the $CS^+/US_{PT}$ trials but also the majority of $CS^-/US_{PT}$ trials were perceived as painful. As we consider the pain threshold in itself to be arbitrary (i.e., either just painful or just not painful), we would expect not more than 50% of $CS^-/US_{PT}$ trials to be rated as painful. The elevated percentage of painful $CS^-/US_{PT}$ trials may indicate a suboptimal calibration of the electrocutaneous stimulus or sensitization throughout the experiment, this should not have affected the overall results of our study.

The present findings bear important clinical implications: it is well established that classical conditioning can drive pain expectancies and pain-related fear, and while such an effect has long been suggested for pain itself the scientific basis for this claim has been lacking. By showing that it is possible to modulate pain threshold ratings through classical conditioning in the laboratory independent of the other pain-related constructs mentioned previously, it becomes more plausible that similar mechanisms occur in chronic pain. Specifically, in the acute pain phase patients may learn to associate the nociceptive input with other concurrent somatosensory stimuli implicitly or explicitly, such that borderline painful stimulation becomes more likely to be perceived as painful thus promoting maladaptive behaviors, cognitions and, eventually, chronicity.

Future studies to replicate and extend our findings about classical conditioning of pain would be useful, as studies on this matter are still rare. Our sample was highly homogenous as it consisted of healthy students, primarily females around 20 years of age. In order to draw any firm conclusions about classical conditioning of chronic pain, replication in other populations that resemble demographics of individuals suffering from chronic pain more closely is necessary. Additionally, the present study focused on discrete locations of stimulation, but it might be valuable to investigate whether the conditioned

shift in report of pain thresholds does also generalize to proximal or distal areas, in order to better understand the spreading of pain. Furthermore, as mentioned previously, it would be interesting to investigate whether the discreteness of the US affects pain conditioning by directly comparing the different types of nociceptive stimulation. Lastly, future studies may include psychophysiological measures in addition to self-reports of pain to get a more complete picture of the effects of pairing noxious and non-noxious stimuli on pain perception and other pain-related outcomes.

## CONCLUSION

This study provides evidence that ratings of at-pain-threshold electrocutaneous stimuli can be modified through classical conditioning. Our results, therefore, advance the debate about whether or not pain itself can be classically conditioned and, although the study was conducted in healthy participants, it contributes to our understanding of the development and maintenance of chronic pain. From a scientific perspective, classical conditioning may pose a useful method for modelling chronic pain in healthy participants. Furthermore, against the backdrop of broad scientific knowledge about classical conditioning in other psychopathologies (*Breivik, 2016*), the present findings suggest that this line of research bears great potential to improve interventions and preventive measures against chronic pain in the future.

## ACKNOWLEDGEMENTS

The results of this study have been presented at the Pain Research Meeting, Antwerp (Belgium), 18–19 September 2017, for which JT received the "best presentation" prize. The authors would like to thank Jeroen Clarysse for his technical support.

### Funding

This research was supported by the "Asthenes" long-term structural funding Methusalem grant by the Flemish Government, Belgium (grant number: 3H150349). This publication was made possible through funding support of the KU Leuven Fund for Fair Open Access. The funders had no role in study design, data collection and analysis, decision to publish, or preparation of the manuscript.

### Grant Disclosures

The following grant information was disclosed by the authors:
The "Asthenes" long-term structural funding Methusalem grant by the Flemish Government, Belgium: 3H150349.
KU Leuven Fund for Fair Open Access.

### Competing Interests

Victoria J. Madden is supported by the National Research Foundation of South Africa, and a developing countries collaborative research grant by the International Association for the Study of Pain (IASP). G. Lorimer Moseley is supported by a Principal Research Fellowship of

the National Health and Medical Research Council of Australia. G. Lorimer Moseley has received support from Pfizer, AIA Australia, Port Adelaide Football Club, Arsenal Football Club. He receives royalties for books on pain and rehabilitation and fees for lectures on pain and rehabilitation. The remaining authors have no competing interests to declare.

## Author Contributions

- Juliane Traxler conceived and designed the experiments, performed the experiments, analyzed the data, contributed reagents/materials/analysis tools, prepared figures and/or tables, authored or reviewed drafts of the paper, approved the final draft.
- Victoria J. Madden conceived and designed the experiments, analyzed the data, contributed reagents/materials/analysis tools, prepared figures and/or tables, authored or reviewed drafts of the paper, approved the final draft.
- G. Lorimer Moseley conceived and designed the experiments, contributed reagents/materials/analysis tools, authored or reviewed drafts of the paper, approved the final draft.
- Johan W.S. Vlaeyen conceived and designed the experiments, analyzed the data, contributed reagents/materials/analysis tools, authored or reviewed drafts of the paper, approved the final draft.

## Human Ethics

The following information was supplied relating to ethical approvals (i.e., approving body and any reference numbers):

The procedures conformed to the Helsinki Declaration and were approved by the Social and Societal Ethics Committee, Faculty of Psychology and Educational Sciences, KU Leuven (approval reference number: G-2016 11 663).

## Data Availability

Open Science Framework: https://osf.io/r45ce/?view_only=33fa09cabb5d4543ab7798fa644ebdcb.

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
