# Peer review of "Modulating pain thresholds through classical conditioning"

_PeerJ, doi:10.7717/peerj.6486_

## Round 0.1 · original submission · Minor Revisions

Both reviewers provide constructive feedback for improving the description of prior findings, description of rationale of study and experimental details, and interpretation of what was conditioned. Both recognize the value of the paper but point out ways you can clarify many points to assist readers to understand what you did and what you found. In particilar, I think it is important to clarify the location of the stimulation on the back in the text, You need to state the type of stimulus more clearly in your summary conclusions because of the conflicting results with thermal vs vibrotactile stimuli, which can be mentioned in the introduction. You do describe the differences between your conflicting studies in the discussion, but it will help the reader to know the stimuli were different in the introduction. It would also help to have at least a brief discussion of the different neural pathways involved in simple touch, vibration, thermal stimulation, and nociception. The reviews are very clear about what each reader wanted clarified methodologically and conceptually, so this can help you improve the paper. In your rebuttal letter please be clear about how you respond to each point of each reviewers.

·

Basic reporting

- The manuscript is very clearly written, and even though the material is quite complex, it was relatively easy to read.
- At several points, the reference style needs to be adjusted. For instance, line 59 or line 145. Avoid nested/double parentheses within references (and in general).
- The introduction provides a nice overview of the literature. When it comes to describing the literature on conditioned allodynia, however, some more information and details would be warranted. For instance, the two studies by Madden et al show conflicting findings, but no explanation is given on why this may be. What could explain the discrepancy? The current study is a follow-up on these studies, but it is not clear to the reader what the current study adds to these two previous studies; how it is different; how the current study will resolve some of the conflicts (if applicable)? I would suggest elaborating a bit more on what gaps there are and how the current study will address these gaps.
- Something that was unexpected when reading the manuscript was the location of the stimuli (both the CS and US). There is no mention of the locations, hence I expected a standard location such as the hand or arm. The authors choose for the back, but this is not explicitly mentioned in the paper, and is only apparent from the figures. I would suggest providing a rationale in the text, and also mention it in the abstract.
- Related, it would be good to mention in the abstract that the CS was presented at a similar location to the US and that the stimuli had the same intensity (but different other properties).
- The rationale for choosing at-pain-threshold stimuli is not clear enough. I believe you choose these stimuli because they are either experienced as painful or non-painful – at chance level (theoretically). This is mentioned in the discussion, but is a crucial aspect of the current study that is not addressed clearly enough. This also explains why you choose the percentage of trials experienced as painful as the primary outcome (correct me if I’m wrong). This should be stated much more clearly and explicitly, because this will not be clear to most readers. I would suggest being more explicit about this aspect, and presenting clear hypotheses regarding the primary outcome.
- The figures are clear and support the main findings.
- Some more specific questions about the introduction:
* When talking about non-nociceptive signaling, do you refer to somatosensory/tactile stimuli or more in general?
* Line 76 – should this be USpt?

Experimental design

- The methods and design are generally sound and well described and fit with the aims and scope of PeerJ.
- As mentioned above, in my opinion the research question and how it fills knowledge gaps could be clearer and more explicit (see Basic Reporting).
- I am impressed with the extent of the pilot testing, and how that has shaped the design further. It is refreshing to read about this part of the study, that is often not mentioned in publications and kept behind closed doors.
- I struggle a bit with the choice of the USpt in relation to the USp and USnp. The way I understand it, the USpt has the same intensity as the USp and USnp, but has a different duration (3 pulses versus 7 or 1). Did the authors verify that indeed USp was experienced as painful and USnp as non-painful during the calibration phase for instance? And how was the USpt rated prior to conditioning? Or in other words: can we assume that they are really experienced the way it is presented (as either painful or non-painful)? And related, could the participants distinguish between these three stimuli?
- Related to the point above, if the participants cannot really distinguish between the stimuli (since they’re the same intensity, on same location where tactile acuity is quite variable and bad) – can you really conclude that pain thresholds are modulated, or would it rather mean that the manipulation (i.e., the conditioning) was successful?
- Some specific questions about the methods:
* For the sample size calculation, one of the papers from Madden et al was chosen (the one that showed a difference). Can the authors explain why? Was this paper more similar to the current study?
* Were any participants excluded because they were unable to discriminate between the vibrotactile stimuli? I could imagine, since tactile acuity on the low back is very variable across participants.
* Line 143: Individual differences – I would suggest choosing a different title here (e.g., Affect) or just name it Positive and Negative Affect
* line 278 – USt should be USpt

Validity of the findings

- Statistics are generally sound, and the authors have looked into several different controls (e.g., effect of contingency awareness, mislocalization of tractor).
- Some specific questions about the results:
* Line 326: change in intensity rating for CS+ was greater than for CS-. It is unclear whether simple effects were tested for this result.
* Fig5 gives intensity ratings for acquisition and test phase, but not for baseline. Could the authors add this information? Can be referred to at line 321.
* The authors sometimes use CS+/USp and sometimes just CS+. I would suggest being consistent in naming (including in figures).
* Line 333-334 – A main effect should not be interpreted in the presence of an interaction effect.
* Table 2 – it is unclear what mA refers to. Also, it is unclear why a table was chosen instead of a figure (as for the other results). I would suggest using a figure to show this data pertaining to the primary analysis.
* Fig5: Was the SPARS rating transformed to another scale (instead of -50 to 50)? Or does this figure show us that all intensities are perceived above 0 and hence as painful? If the latter would be the case, that may be a problem.
* Recalibration – Although I understand why you choose to recalibrate, it would be good to show some data on how the pain thresholds changed over time. The authors mention the recalibrated values were no different across CS+ and CS-, but it would be nice to see some data on how much they changed.
- Even though I agree that this is an important question and has relevance for chronic pain, I do not agree that the present study investigates ‘whether classical conditioning could be one of the mechanisms underlying the development and maintenance of chronic pain’ (line386-387). This is too strong, and I would suggest rephrasing this.
- When thinking of how this is relevant for chronic pain, I miss some discussion on what this means and how this would actually contribute to development and maintenance of chronic pain, above and beyond other pain-related constructs that can be conditioned (e.g., fear of pain). Related, what mechanisms could be involved in this phenomenon?
- General remark: the authors call it conditioned allodynia, but I am not sure whether this is accurate when assessing pain thresholds (that are theoretically at the border of non-painful and painful).

Additional comments

Overall
The authors present a well-designed study asking the question whether pain itself can be conditioned. Previous studies have shown that all sorts of pain-related constructs can be conditioned, but less is known about pain itself. There has been some work on this topic with conflicting results; some from the same group, which has been a basis for the current study. I believe the topic and question are highly interesting and very relevant for the field of pain and chronic pain. The question has been systematically addressed, well described, and results are quite clear.

·

Basic reporting

The present manuscript uses a clear und professional English language. It terms of the used language the manuscript is well-written und easy to follow. However, due to the structure, particularly in the methods section, the manuscript is in part confusing, because it refers to methodological aspects before they are actually explained. For example, in the section “Stimuli”, line 119 it is mentioned that a procedural adaption was made “to account for perceived control…”. However, why participants should have the impression of perceived control is explained much later in the section “Procedure”. This happens in a couple of instances. I would strongly recommend to explain the procedure much earlier in the methods section, followed by all the details in the sections “Stimuli” and “Measure and Outcomes”, to make it easier to follow the descriptions.

Although the introduction is well-written and refers the relevant literature, it is imprecise when it comes to the description of what should be conditioned. This means, the authors state that “pain itself can occur as a CR”, but “pain itself” is very vague because pain comprises very many very different aspects from pure nociceptive signaling, to the conscious perception of pain with several aspects, and motor responses, autonomic responses etc. It should be clearly stated what aspect of the pain experience is supposed to be the CR. In addition, it would be very helpful if also the CS, US, UR are precisely named. Based on these clarifications the discussion could also be adapted. In addition, the authors should clearly refer to classical conditioning, because in the context of operant conditioning several attempts were made to condition perceptual processes and underlying neural correlates.

Figure 3 and 4 appear somewhat redundant. I’d suggest omitting Figure 3 because all relevant information is contained in Figure 4 as well.

Experimental design

The reported research is within the scope of the journal, filling a current gap in the literature in psychological pain research.

Although the research question is well-defined and accordingly introduced in the introduction, I think it need a bit more precision as described above, i.e. being more precise about the CR because “pain itself” is a ambiguous term. The research question tackles a long-standing discussion on whether conditioning/learning can alter only processes related to pain such as fear or avoidance behavior or whether the perception of the pain/underlying nociceptive signaling can be altered by classical conditioning. Since classical conditioning plays a very important role in the context of pain and pain becoming chronic, this research question is highly relevant.

The authors used a sophisticated method to induce classical conditioning of pain perception. The methods are adequate and described in a manner, which would allow replication. The only part that I did not understand was how the readjustment by the participants of the applied stimuli between blocks (and phases) of the experiment were used for adjusting the applied stimulus intensities. It remained unclear to me (possibly due to the structure of the methods sections, see above) whether the stimulus intensities were actually adjusted to the intensities participants adjusted themselves.
Technical and ethical standards are kept.

Validity of the findings

The analysis is sound and the results appear robust. The statistics are appropriate and sufficiently. Only the descriptions of the main hypothesis, section “Effects on Pain thresholds” need some more explanation. The percentages when the stimulation was rated as painful are given, but because percentages can be misleading absolute numbers (in the manner: xy number of trials out of xy) are necessary.

For the secondary analysis reported in section “Effects on pain intensity”, the values reported in the second paragraph, lines 365-371, do not fit the values reported in Figure 5. In Figure 5, on the y-axis the values shown do not fit the descriptions of the SPARS rating scale given in the methods section.

Although the results are interesting and convincing, I have a conceptual issue with the reporting of the result and the interpretation given by the authors. The authors state that they aimed at modulating pain thresholds by classical conditioning. However, with the procedure they used in combination with the results reported, make it for me hard to understand, how the conditioning should have affected the pain thresholds directly. By the conditioning procedure, pain perception was modulated in that stimuli of equal intensity were more often rated as painful when combined with the CS+, compared to when combined with the CS-. This should in the end increase perceived pain, resulting as a consequence in a modulation of pain thresholds, but the CR appears not to have been the pain threshold itself, making the interpretation of the results complicated. I’d recommend sticking with a technical description of what was modulated precisely with the procedure and describing in the results section the given ratings (painful vs non-painful) instead of referring to pain thresholds. To get a more precise idea of how and if at all the thresholds were condition, the authors should add more information on the self-adjusted stimulation intensities by the participants, in particular when doing this in reference to the CS+ and CS- separately at the end of the procedure. Here, it is only mentioned in the discussion that these self-adjusted intensities did not differ, but I could not find data on this in the results section.

Table 2 is confusion as the title says “Mean stimulus intensity ratings” but in the table values in mA are given.

To understand the procedure better, a rational should be given, why a self-adjustment procedure war used for the calibration of stimulus intensities, while all other stimuli were applied without self-adjustment and thus control by the participants.

Additional comments

Line 325-326: it is unclear what is meant by “change in intensity ratings”. Please explain.

When describing the procedure in the discussion, the authors describe the procedure as simultaneous classical conditioning. I’d recommend to use the term “delay conditioning” as a commonly used description of such a procedure.

Instead of citing a doctoral thesis (Bräscher 2014), I recommend citing the paper: Bräscher, A. K., Kleinböhl, D., Hölzl, R., & Becker, S. (2017). Differential classical conditioning of the nocebo effect: increasing heat-pain perception without verbal suggestions. Frontiers in psychology, 8, 2163.

---

## Round 0.2 · accepted · Accept

Your responses to the comments of the previous review were clear and thorough. I am sure you agree with me that the paper is now improved. The paper is a valuable contribution to the literature.

#